# Generalized Hierarchical Co-Saliency Learning for Label-Efficient Tracking

**DOI:** 10.3390/s25154691

**Published:** 2025-07-29

**Authors:** Jie Zhao, Ying Gao, Chunjuan Bo, Dong Wang

**Affiliations:** 1Dalian University of Technology, Dalian 116024, China; zj982853200@dlut.edu.cn (J.Z.); wdice@dlut.edu.cn (D.W.); 2Henan Key Laboratory of Safety Technology for Water Conservancy Project, Henan Water & Power Engineering Consulting Co., Ltd., Zhengzhou 451450, China; 3Dalian Minzu University, Dalian 116620, China; bcj@dlnu.edu.cn

**Keywords:** visual tracking, weakly supervised learning, co-saliency attention, egocentric tracking

## Abstract

Visual object tracking is one of the core techniques in human-centered artificial intelligence, which is very useful for human–machine interaction. State-of-the-art tracking methods have shown their robustness and accuracy on many challenges. However, a large amount of videos with precisely dense annotations are required for fully supervised training of their models. Considering that annotating videos frame-by-frame is a labor- and time-consuming workload, reducing the reliance on manual annotations during the tracking models’ training is an important problem to be resolved. To make a trade-off between the annotating costs and the tracking performance, we propose a weakly supervised tracking method based on co-saliency learning, which can be flexibly integrated into various tracking frameworks to reduce annotation costs and further enhance the target representation in current search images. Since our method enables the model to explore valuable visual information from unlabeled frames, and calculate co-salient attention maps based on multiple frames, our weakly supervised methods can obtain competitive performance compared to fully supervised baseline trackers, using only 3.33% of manual annotations. We integrate our method into two CNN-based trackers and a Transformer-based tracker; extensive experiments on four general tracking benchmarks demonstrate the effectiveness of our method. Furthermore, we also demonstrate the advantages of our method on egocentric tracking task; our weakly supervised method obtains 0.538 success on TREK-150, which is superior to prior state-of-the-art fully supervised tracker by 7.7%.

## 1. Introduction

Visual object tracking (especially egocentric tracking) [1,2,3] is one of the core techniques in human-centered artificial intelligence [4]. Tracking any object of interest is very useful for human–machine interaction, which has many realistic applications (virtual reality, surveillance, vision control, to name a few). It has been confirmed that deeper neural networks are important to obtain better tracking performance, including convolutional neural networks (CNN) [5] and the recent Transformer networks [6]. Most of these deep network-based trackers are data-driven and fully supervised methods, since their training processes require plenty of training data with precise annotations. Some large-scale tracking datasets are developed to satisfy the training requirements for deep tracking models. For instance, LaSOT [7] and GOT-10k [8] provide over 2.8 M and 1.4 M manual annotations for training, respectively. Each of their sequences is annotated and checked frame-by-frame by workers. However, such manually annotating videos frame-by-frame is quite time- and labor-consuming. It is a huge workload for larger video datasets collected on the Internet, where a steady stream of new videos will be uploaded every day. Therefore, label-efficient tracking is a problem worthy of studying, which focuses on effectively reducing requirements for precise annotations when training deep models, and enabling the model to make full use of unlabeled frames.

Despite being an important task, there are only few works exploring label-efficient tracking. Most of them attempt to investigate unsupervised tracking frameworks [9,10,11] without utilizing any annotation. Due to the lack of guidance for precise states of the specific target, these unsupervised trackers still have performance gaps with their baseline supervised methods. Further, Liu et al. [12] propose a reliable sample selection strategy for weakly supervised tracking, where poor-quality samples can be filtered by the proposed strategy. Approximately 9.13% manual annotations are required for training their models, and the obtained performance is close to 80% of full supervision. Different from these methods, in this paper, we propose a generalized co-saliency attention mechanism, which can be hierarchically integrated into various tracking frameworks to achieve weakly supervised tracking. Experiments demonstrate that our models require 3.33% annotations to achieve over 95% performance of full supervision.

The attention mechanism is a common and effective strategy for optimizing feature representations in many computer vision tasks. In [13], an attention-based selection mechanism is proposed to achieve feature filtering and fusion for salient detection. Liu et al. [14] propose a co-saliency attention mechanism to optimize the accuracy of person re-identification. Different from the motivations of these methods, we propose a co-saliency attention (CSA) module, which can mine rich co-salient visual information from unlabeled frames from both spatial and channel levels, thereby achieving weakly supervised tracking. The obtained co-salient attention map for the current search feature can enhance the common salient foreground information while weakening the background area. The proposed CSA can be treated as a generalized module that can be flexibly integrated into various tracking frameworks, achieving weakly supervised tracking and keeping the competitive tracking performance. Considering that the appearance and location of the target in a short clip have small changes, we adopt sparsely labeled datasets to train the model in a weakly supervised manner, that is, manually annotating one bounding box of the target every 30 frames (around one second for most videos). The remaining unlabeled frames will be arranged pseudo-labels by linear interpolation according to the first and last labels of the fragments. During training, 3.3% frames with precise labels will be sampled as template and search images, which are involved in the calculation of tracking losses, while those 96.7% unlabeled frames will be used as part of the input of our CSA module to assist in improving the current search features. This operation enables the model to make full use of the effective target information of unlabeled frames and save nearly 30 times the manual annotation costs.

As concluded in [15], human-centric consumer applications play a critical role in improving the quality of life. Due to the importance of these first-person vision tasks, egocentric tracking [1,2] has gradually obtained more and more attention in recent years. Compared with general VOT with third-person vision, challenges of egocentric tracking are reflected in rapid appearance changes, large camera motions, occlusions caused by frequent hand interactions, and so on. Most prior tracking methods have limited performance when facing this kind of sequences due to considering only the match between the initial frame and the current frame. In contrast, our CSA module enables the model to refer to rich visual information from multiple frames, and enhance the target representation of the current frame by calculating co-saliency attention maps. Therefore, our weakly supervised method also provides a solution for egocentric tracking.

We hierarchically integrate our CSA into two CNN-based tracking models (anchor-based method SiamRPN++ [5] and anchor-free method Ocean [16]), and a Transformer-based tracking framework (OSTrack [6]). Extensive experiments on four general VOT datasets and an egocentric tracking dataset can demonstrate the effectiveness and generalization of our weakly supervised tracking method on both third-person vision and first-person vision. An intuitive comparison between our weakly supervised methods and corresponding fully supervised baseline trackers is illustrated in Figure 1, emphasizing the balance between annotation efficiency and tracking performance on LaSOT.

Main contributions of our work are summarized as follows:We propose a generalized co-saliency attention mechanism, which can be integrated into various tracking frameworks to achieve weakly supervised tracking. Valuable visual information from unlabeled frames can be explored by our method, which reduces plenty of manual annotation costs during training.We record historical visual features and explore their co-salient information via our CSA in the inference. This enables the model to be more robust to challenges like appearance changes and occlusions, which provide a solution for egocentric tracking.We integrate our CSA module into two CNN-based methods, and a Transformer-based method. Extensive experiments on four general VOT datasets and an egocentric tracking dataset demonstrate that our weakly supervised tracking models can achieve over 95% performance of full supervision, using only 3.33% manual annotations.

## 2. Related Work

### 2.1. Supervised Visual Tracking

Existing deep learning-based tracking methods can be divided into two categories in terms of the type of backbone networks, which are CNN-based trackers and Transformer-based trackers, respectively. Some representative CNN-based methods include Siamese-based trackers [5,16,17,18,19] and discriminative methods [20,21,22,23,24,25]. Due to the powerful global representation capability of Transformer networks, the recent state-of-the-art (SOTA) tracking methods are gradually transitioning to employ Transformer-based backbone networks. To give full play to the global advantages of Transformer networks, one-stream tracking frameworks based on pure Transformer architectures are proposed, such as OSTrack [6] and MixFormer [26]. These methods adopt the pretrained ViT [27] to merge the feature extraction and feature fusion, which achieves the deep fusion between template and search images. Generalized representations by pretrained models also boost tracking results. Although aforementioned SOTA trackers perform well on many tracking scenes, their models require a large amount of manually annotated video data during the fully supervised training process. Annotating videos frame-by-frame is inevitably a labor- and time-consuming task.

### 2.2. Label-Efficient Visual Tracking

Regarding label-efficient visual tracking, some works aim to achieve unsupervised tracking. UDT [9] proposes an unsupervised tracker trained on large-scale unlabeled videos. This method learns to locate targets by repeatedly performing forward tracking and backward verification. PUL [10] proposes a progressive unsupervised tracking framework. A background discrimination model based on contrastive learning is employed to progressively mine temporal corresponding patches. A noise-robust loss function is further proposed to mitigate the effect from noisy samples. A crop-transform-paste operation is developed in [11], which enriches training data by simulating appearance variations. In its self-supervised tracking setting, only one annotated frame per video is required. Unlike these methods that rely on complex frameworks incompatible with existing advanced trackers, our CSA module can be easily integrated into varying supervised tracking pipelines to enable weakly supervised tracking, offering greater flexibility and practicality. Moreover, due to the lack of precise annotations to guide the training process, the tracking performance of these unsupervised methods still has gaps with supervised trackers.

To make a trade-off between the performance and annotation costs, Liu et al. [12] propose a reliable sample selection strategy for weakly supervised tracking. Given the initial annotation, a correlation filter-based tracker using hand-craft features is employed to generate pseudo-labels for subsequent frames. These pseudo-labels are then used to crop training samples for Siamese-based trackers. Poor-quality samples assessed by the proposed strategy will be filtered. Approximately 9.13% manual annotations are required in this work for weakly supervised training, and the obtained performance is close to 80% of full supervision. This method focuses solely on generating accurate training labels to enable weakly supervised training, with performance upper-bounded by that of fully supervised models. In contrast, our approach integrates the proposed CSA module into existing trackers to enhance the model itself via co-saliency learning, allowing it to surpass the fully supervised performance in some cases. Experiments show that our method can obtain competitive performance with their baseline trackers, using only 3.33% manual annotations for training.

### 2.3. Egocentric Visual Tracking

Since egocentric vision is closer to human perceptions of the real world, it is important for various real-world applications, such as virtual reality [28], surveillance [29], and vision control [30]. As one of the fundamental tasks for egocentric vision, visual object tracking on egocentric sequences has begun to be explored by more and more researchers. Dunnhofer et al. [1] propose an egocentric tracking benchmark, TREK-150. This benchmark consists of 150 densely annotated videos under kitchen scenes. EgoTracks dataset [2] has recently been proposed to further support the development of egocentric tracking tasks. This dataset sources videos from Ego4D [4] and supplements dense annotations for both training and testing subsets.

Compared with the general tracking task (third-person vision), egocentric tracking is more challenging due to rapid appearance changes, large camera motions, occlusions caused by frequent hand interactions, and so on. Since our method can learn co-salient foreground information from multiple frames. The recorded rich appearance changes enable our model to better adapt to egocentric tracking. To demonstrate this conjecture, we additionally evaluate our method on egocentric tracking dataset TREK-150. More comparisons and analysis can be found in Section 4.3.

## 3. Approach

Algorithm 1 summarizes the weakly supervised training pipeline of our CSA-enhanced tracker. Training videos are segmented using sparse annotations, and missing labels are interpolated. Given a baseline tracker, our CSA modules are integrated into it to achieve weakly supervised training.
**Algorithm 1** Weakly Supervised Training of CSA-Enhanced Tracker
  1:**Input:** Training videos *V*, sparse annotations *A*, baseline tracker *T*;  2:**Output:** CSA-enhanced Tracker $T_{CSA}$ with weakly supervised training;  3:**Segment** each video of *V* into short fragments based on sparse annotations *A*;  4:**Interpolate** missing annotations for unlabeled frames using linear interpolation;  5:**Integrate** CSA modules into the baseline tracker *T* to build $T_{CSA}$;  6:**Train** $T_{CSA}$ in a weakly supervised manner:
    • Sample *template* and *search image* pairs from annotated frames;    • Sample *intermediate images* from unannotated frames;  7:**return** CSA-enhanced Tracker $T_{CSA}$ with weakly supervised training.

In this section, we first introduce our weakly supervised module based on co-saliency learning in Section 3.1. Then, we describe our weakly supervised data processing in Section 3.2. In the end, we present how we integrate our method into existing tracking frameworks to achieve weakly supervised tracking in Section 3.3 (CNN-based framework) and Section 3.4 (Transformer-based framework), respectively.

### 3.1. Weakly Supervised Co-Saliency Learning

Co-saliency learning aims to extract the common significant foreground areas from a set of images. Inspired by the prior work [14], we propose a co-saliency attention mechanism (CSA) to construct a weakly supervised tracking environment. Figure 2 shows the detailed architecture of the proposed CSA module. The input of the module contains feature maps of multiple frames from the same video, denoted as ${\left\{f_{t}\right\}}_{t=1}^{T}$. *T* represents the total number of feature maps and $f_{t}\in R^{C\times H\times W}$ has H×W resolution with *C* dimensions. Our CSA module consists of two branches, i.e., spatial branch and channel branch. In doing so, the common foreground information of multiple input frames can be obtained considering both spatial positions and common important channels. Specifically, to reduce computations and implement our CSA as a lightweight module, all of the input feature maps first go through the channel and spatial dimension reduction layers (shown as Channel Conv and Spatial Conv in Figure 2). These two dimension reduction layers are both implemented by a convolutional layer followed with ReLU, and output two types of feature maps with low dimensions $f_{t}^{c}\in R^{C\times H_{L}\times W_{L}}$ and $f_{t}^{s}\in R^{C_{L}\times H\times W}$, where $C_{L}$, $H_{L}$ and $W_{L}$ are much smaller than *C*, *H*, and *W*. Then, we employ the normalized cross correlation operation (NCC) to calculate the similarity among multiple frames. The NCC operation of two feature vectors with dimension $C_{c}$ can be formulated as(1)$$NCC\left(P,Q\right)=\frac{1}{C_{c}}\frac{\sum_{g=1}^{C_{c}}\left(P_{g}-\mu_{P}\right)\left(Q_{g}-\mu_{Q}\right)}{\sigma_{P}\times \sigma_{Q}},$$where $\left(\mu_{P},\mu_{Q}\right)$ and $\left(\sigma_{P},\sigma_{Q}\right)$ represent the couples of the mean and standard deviation of the feature vectors *P* and *Q*, respectively. The aforementioned two types of low-dimension feature map sets ${\left\{f_{t}^{c}\right\}}_{t=1}^{T}$ and ${\left\{f_{t}^{s}\right\}}_{t=1}^{T}$ are, respectively, input to the channel NCC block and spatial NCC block to calculate the co-salient information of all *T* frames in channel and spatial levels.

In the spatial NCC block, every spatial location (i,j)(1≤i≤H,1≤j≤W) of the feature map $f_{t}^{s}$ with dimension $C_{L}\times H\times W$ can be viewed as a $C_{L}$ dimensional feature vector (denoted as $f_{t}^{s}\left(i,j\right)$) that describes the local information at (i,j). We employ NCC to calculate the similarity of the information in the frame’s position (i,j) with the information in all positions of other frames, formulated as(2)$$\begin{array}{cc} SA_{t}^{i,j} & =NCC(f_{t}^{s}\left(i,j\right),f_{k}^{s}\left(h,w\right)),\\ 1 & \leq k\leq T,k\neq t\\ 1 & \leq h\leq H\\ 1 & \leq w\leq W. \end{array}$$

The output of the spatial NCC block is a 3D attention map $SA_{t}\in R^{(T-1)HW\times H\times W}$. It records the correlation information between each position of the frame *t* and all positions of other (T−1) frames. Thus, spatial locations of the common foreground consulting with multiple frames are activated.

The process of the channel NCC block is similar to the spatial NCC block. First, the feature map $f_{t}^{c}$ with dimension $C\times H_{L}\times W_{L}$ is flatted to 2D matrices. It represents that every channel in the feature can be viewed as a $H_{L}W_{L}$ dimensional vector $f_{t}^{c}\left(l\right)$. The channel NCC block computes the channel attention map $CA_{t}\in R^{(T-1)C\times C}$, which records the similarity between each channel of the frame *t* and all channels of other (T−1) frames. The common important channels of all video frames can then be enhanced. This process can be formulated as(3)$$\begin{array}{cc} CA_{t}^{l} & =NCC(f_{t}^{c}\left(l\right),f_{k}^{c}\left(e\right)),\\ 1 & \leq k\leq T,k\neq t\\ 1 & \leq e\leq C. \end{array}$$

The two attention maps recording the similarity information, i.e., SA and CA, are then summarized by a convolutional layer, and transformed to $Z_{s}\in R^{1\times H\times W}$ and $Z_{c}\in R^{C\times 1\times 1}$, respectively. The final spatial-channel attention map is formulated as follows:(4)$$Attention=sigmoid(Z_{c}⊗Z_{s}).$$

Among them, Attention is a C×H×W matrix, where common important channels and the co-salient spatial positions are highlighted with reference to multiple video frames, while other information will be weakened. In this paper, we calculate the co-saliency attention map of the current frame’s feature map, with reference to the information from the other two frames, including the initial template and a random intermediate frame. The attention map is multiplied with the feature map of the current frame, making the target position and important channels more salient. Meanwhile, the background information and other trivial channels of the final feature map of the current frame are suppressed.

### 3.2. Weakly Supervised Data Processing

Since the target positions and shapes usually do not change much in a 30-frame clip, which occurs in a second for most videos, we adopt sparsely labeled training datasets to achieve weakly supervised tracking. That is, labeling one bounding box of the target every 30 frames. The input images to CSA are used to generate co-saliency attention maps, which do not require precise target annotations but only need the target to be present within the search regions cropped based on pseudo-labels. As a result, CSA inputs from unlabeled frames are highly tolerant to pseudo-label inaccuracies. Therefore, in this work, we employ linear interpolation to generate pseudo-labels for unlabeled frames, illustrated as Figure 3.

Although linear interpolation is not reliable for modeling complex scenarios such as nonlinear target motion, it is sufficient for our method because precise annotations are not strictly required. To support this claim, we evaluate the quality of the pseudo-labels generated on LaSOT dataset using two metrics. As shown in Table 1, we report results under varying annotation sparsity. “IoU” denotes the Intersection over Union between pseudo-labels and groundtruth annotations, while “Target ratio” measures the proportion of the target included in the search region cropped based on the pseudo-label. We observe that while the IoU between pseudo-labels and groundtruth drops significantly with increased sparsity (e.g., from 0.679 at sparsity 30 to 0.445 at sparsity 150), the target ratio remains consistently high. Even at a sparsity level of 150, the target ratio exceeds 0.9, indicating that the cropped search regions still contain nearly complete target information. This property fits well with our method, as the CSA module only requires the target to be present in the search region, not precisely localized. Thus, pseudo-labels from linear interpolation are sufficiently reliable, enabling effective weakly supervised training.

### 3.3. CNN-Based Weakly Supervised Tracking

Taking the baseline tracker SiamRPN++ [5] as an example, we present how we hierarchically integrate our CSA module into a CNN-based tracking framework to achieve weakly supervised training, shown as Figure 4.

For the baseline tracker SiamRPN++, its inputs consist of an initial template image *z* and a current search image $x_{c}$. A CNN-based backbone network is employed to extract their multi-level feature maps, and then places them into corresponding region proposal networks (RPN blocks in Figure 4). After fusing the outputs from multiple RPN blocks, the classification and regression branches will give a dense prediction of the target. The former is in charge of choosing the best proposal through classifying the foreground and background, while the latter is used to refine the proposal and estimate the offsets of anchor boxes. This process can be formulated as(5)$$\begin{array}{cc} A_{w\times h\times 2k}^{cls} & ={\left[\varphi \left(x_{c}\right)\right]}_{cls}⋆{\left[\varphi \left(z\right)\right]}_{cls}\\ A_{w\times h\times 4k}^{reg} & ={\left[\varphi \left(x_{c}\right)\right]}_{reg}⋆{\left[\varphi \left(z\right)\right]}_{reg}, \end{array}$$
where φ(·) represents the feature maps extracted from the backbone network, *k* represents the number of predefined anchors, and ⋆ is a convolution operation on feature maps of search regions, with the template features as the convolution kernel.

Different from the baseline, there are three different search regions in the inputs of our model, which are cropped from the initial frame, an unlabeled intermediate frame, and the current search frame, respectively, namely $x_{t}$, $x_{m}$, and $x_{c}$. It is worth noting that intermediate frames are cropped based on generated pseudo-labels. Each of these three search regions has a shared search branch to extract feature maps. As shown in Figure 4, multiple of our CSA modules are hierarchically inserted after different layers of the backbone network. The features of the corresponding layers output by the three search branches are input to our CSA modules. For the current search branch, the input of the next backbone layer will be replaced by the output of the corresponding CSA module, i.e., the enhanced feature maps of the current search image. After integrating our CSAs, the tracking process can be reformulated as(6)$$\begin{array}{cc} A_{w\times h\times 2k}^{cls} & ={\left[CSA\left(\varphi \left(x_{t}\right),\varphi \left(x_{m}\right),\varphi \left(x_{c}\right)\right)\right]}_{cls}⋆{\left[\varphi \left(z\right)\right]}_{cls}\\ A_{w\times h\times 4k}^{reg} & ={\left[CSA\left(\varphi \left(x_{t}\right),\varphi \left(x_{m}\right),\varphi \left(x_{c}\right)\right)\right]}_{reg}⋆{\left[\varphi \left(z\right)\right]}_{reg}. \end{array}$$

We can find that after integrating our CSA modules, the tracking framework is able to use both labeled (i.e., the initial and current search frames) and unlabeled data (i.e., intermediate frames), therefore constructing a weakly supervised training environment. The precisely labeled data can provide accurate target information and guide the model optimization by calculating precise losses, while the unlabeled data is mainly utilized to calculate the co-salient information of multiple frames, and then enhance the current search features.

### 3.4. Transformer-Based Weakly Supervised Tracking

Due to their powerful global representation ability, Transformer-based tracking methods have gradually become the mainstream. To demonstrate the generalization ability of our method on Transformer-based tracking frameworks, we choose a one-stream Transformer-based tracker OSTrack [6] as the baseline, and integrate our CSA modules into its framework. In OSTrack, the template and search region are first split to multiple patches with a fixed size. Each patch can be treated as a processing unit in the Transformer network, namely token. Passing through a patch embedding layer, template tokens and search tokens are concatenated together and iteratively input into multiple Transformer blocks (TF blocks), shown as the top row in Figure 5. After executing self-attentions and cross-attentions in these TF blocks, the search tokens output from the last TF block are sent to a head module to predict the target state.

Similar to CNN-based integration, we employ three shared backbone branches to encode three types of tokens along with the template tokens, including initial tokens, intermediate tokens, and search tokens. These tokens are processed through a series of TF blocks, and three CSA modules are integrated after specific TF blocks to facilitate information interaction and enhancement among the three branches. Specifically, the output tokens from the current TF blocks in each of the three branches, including initial tokens, intermediate tokens, and search tokens, are reshaped into feature maps. These feature maps are then passed to the corresponding CSA modules for co-saliency calculation. The CSA module enhances the search tokens by incorporating additional context and information from the other branches. The enhanced search tokens will then replace the original search tokens, continuing the Transformer encoding process with the updated tokens, as shown by the orange tokens in Figure 5.

## 4. Experiments

Our work is implemented in Python using PyTorch. Training is performed using NVIDIA A800 GPUs, Santa Clara, CA, USA, while inference is conducted on a single NVIDIA GeForce RTX 2080 Ti GPU, Santa Clara, CA, USA. To show the generalization ability of our weakly supervised CSA module, we choose two classic CNN-based trackers (i.e., an anchor-based tracker SiamRPN++ [5] and an anchor-free tracker Ocean [16]) and a Transformer-based tracker OSTrack [6] as baselines. We transform them into weakly supervised tracking algorithms by integrating our CSA module, denoted as SiamRPN++_CSA_, ${\mathrm{Ocean}}_{\mathrm{CSA}}$, and ${\mathrm{OSTrack}}_{\mathrm{CSA}}$, respectively. We compare the tracking performance of our weakly supervised methods with the three baseline trackers and other trackers on four general VOT benchmarks, including LaSOT [7], GOT-10k [8], OTB2015 [31], and UAV123 [32]. In addition, to further demonstrate the effectiveness of our method on egocentric tracking, we also evaluate our method ${\mathrm{OSTrack}}_{\mathrm{CSA}}$ on an egocentric tracking benchmark, TREK-150 [1].

### 4.1. Implementation Details

**Network archtecture.** All of our methods follows the same input size as their baselines. For SiamRPN++_CSA_ and ${\mathrm{Ocean}}_{\mathrm{CSA}}$, the sizes of the input template images and search regions are set to 127×127 and 255×255, while for ${\mathrm{OSTrack}}_{\mathrm{CSA}}$, they are set to 128×128 and 256×256. Regarding our CSA module, the spatial dimension reduction layer is implemented by a convolutional layer where the kernel size is 1. Its output dimension is set to 128 for CNN and 256 for Transformer. Meanwhile, the channel dimension reduction layer is a convolutional layer where the kernel size is 7, the stride is 2, and the output dimension is the same as the input. For CNN-based methods, we choose ResNet50 [33] as the backbone, and CSA modules are integrated on the last three layers (since Ocean does not apply multi-layer features to calculate cross-correlation, we only set up one CSA module on its last layer). For Transformer-based methods, we choose ViT-B [27] as the backbone, and similarly set up our CSA modules on the last three Transformer blocks.

**Training.** We follow the same training settings of corresponding baseline trackers. For both two CNN-based methods, we use synchronized stochastic gradient descent (SGD) over 8 GPUs where the batch size of each GPU is 16. GOT-10k [8], LaSOT [7], ImageNet VID [34], and TrackingNet [35] are applied to train our CNN-based models. For Transformer-based method, we use AdamW optimizer over 4 GPUs where batch size of each GPU is 32. We follow the three video datasets in the baseline training dataset settings to train our model, i.e., GOT-10k, LaSOT, and TrackingNet. It is worth noting that we use sparse annotations for all training datasets, i.e., using one annotation per 30 frames. Other unlabeled frames are arranged pseudo-labels and sampled only as intermediate frames without involving tracking loss calculation.

**Inference.** In the inference, we set up a memory pool to record feature maps of historical search regions. It consists of the feature map of the initial frame’s search region, and a real-time updated feature of an historical frame with the highest confidence score. This confidence score considers both the confidence of the tracking result and the time factor. It can be formulated as(7)$$S_{penalty}^{t}={\mathrm{Conf}}_{t}+\frac{\mathrm{lg}N_{t}}{1000},$$
where ${\mathrm{Conf}}_{t}$ and $N_{t}$ represent the confidence score of the tracking result and the frame number, respectively. It is worth noting that ${\mathrm{Conf}}_{t}$ is generated from tracking models along with the tracking results.

### 4.2. Performance Comparison on General VOT

We compare our three weakly supervised methods with fully supervised baselines and other trackers on four general VOT benchmarks, i.e., LaSOT, OTB2015, UAV123, and GOT-10k. Moreover, we also compare with existing weakly supervised and unsupervised trackers on OTB2015 and UAV123.

**LaSOT.** It is a large-scale and long-term tracking dataset containing 1120 training sequences and 280 testing sequences with 70 object categories. The average sequence length is more than 2500 frames and there are more than 3.5 M frames in total. Sequences in LaSOT often occur in the situation that targets disappear and re-appear in the view, which better reflects the practical performance of trackers in the wild. The benchmark uses precision (Pre.), normalized precision (Norm Pre.), and success (Succ.) following one-pass evaluation (OPE) to measure the performance of trackers. The precision compares the distance between the tracking results and groundtruth in pixels to measure center location error. To make the evaluation robust to the target size and image resolution, the normalized precision is used to rank trackers using Area Under Curve (AUC) between 0 to 0.5. In addition, the success calculates the Intersection over Union (IoU) between the tracking results and groundtruth to reflect the accuracy of the target’s position and scale. Figure 6 shows the tracking results of our three weakly supervised methods (denoted as the subscript “CSA”) and 10 fully supervised trackers, including SiamFC [17], MDNet [20], ATOM [22], SiamBAN [36], DaSiamRPN [37], SiamFC++ [38], DiMP [21], SiamRPN++ [5], Ocean [16], and OSTrack [6]. Our weakly supervised methods achieve competitive performance to SOTA fully supervised trackers. Using only 3.33% manual annotations in training, our CSA can further improve three baseline trackers by averaged 2.5% in terms of Succ.

**OTB2015**. It is a classic short-term tracking dataset that contains 100 sequences. Two metrics, precision and success, are used for evaluation. As shown in Table 2, our method, OSTrack_CSA_, improves its fully supervised baseline by 1.7% Succ. and 3.1% Pre. In addition, our methods achieve comparable tracking performance with fully supervised trackers, and are superior to existing weakly supervised trackers. Regarding unsupervised methods, we can find that they still have some performance gaps with SOTA trackers.

**UAV123**. Unlike the above datasets, it captures 123 videos via drone aerial photography with more than 110,000 frames. It also uses the precision and success metrics to evaluate the tracking results. As shown in Table 2, our CSA boosts three baseline trackers by 2.0% Pre. on average, using only one-thirtieth of manual annotations.

**GOT-10k.** It contains over 10,000 video segments with 563 object classes and 87 motion forms for short-term tracking. The testing set consists of 180 videos with 150 object classes and 15 motion classes. Different from LaSOT dataset, it introduces average overlap (AO) and success rate (SR) as indicators. The former denotes the average of overlaps between the estimated bounding boxes and groundtruth, while the latter measures the percentage of successfully tracked frames where the overlaps exceed a threshold. The threshold is set to 0.5 and 0.75, and the corresponding indicators are denoted as ${\mathrm{SR}}_{0.5}$ and ${\mathrm{SR}}_{0.75}$, respectively. We compare our methods with 10 fully supervised trackers, as shown in Table 3. Our CSA yields SiamRPN++ 3.7% and 5.6% improvements in terms of the AO and ${\mathrm{SR}}_{0.5}$. Meanwhile, Ocean_CSA increases ${\mathrm{SR}}_{0.75}$ by 4.2%. Regarding the speed, since we record features of historical frames by a memory pool in the inference, tracking speeds of our methods are not sacrificed much, which can still achieve real-time tracking.

### 4.3. Performance Comparison on Egocentric Tracking

To demonstrate our conjecture that our CSA can enable the tracking model to better adapt to egocentric tracking, we evaluate our superior method OSTrack_CSA_ on the egocentric tracking benchmark, TREK-150 (Figure 7) [1]. It consists of 150 sequences in kitchen scenes. All of them are shot under first-person vision. As shown in Table 4, we present the performance of the top ten methods evaluated in its official paper, and the baseline OSTrack. It is worth noting that all of the presented methods are fully supervised trackers except for our method OSTrack_CSA_. It shows that using only 3.33% of manual annotations for training, our method obtains significant performance improvement by 3.6% Succ., and is superior than the prior SOTA method LTMU-H by 7.7% Succ.

In addition, we illustrate qualitative comparisons of our weakly supervised method with its fully supervised baseline tracker on TREK-150. We can find that our method can better address difficult challenges, such as scale variations, appearance changes, partial occlusions, and fast motion.

### 4.4. Ablation Study

To deeply analyze our method from different aspects, we execute a set of ablation studies on LaSOT.

**Impact of CSA**. To demonstrate the effectiveness of our CSA, we present tracking results of three variants of baseline trackers Ocean and OSTrack, shown as Table 5. Among them, “Fully S.” represents the original fully supervised baseline, “Weakly S.” represents using 3.33% manual annotations to train the baseline model, and “CSA” represents our weakly supervised method. The tracking performance inevitably decreases when reducing the number of annotations for training, but this degradation can be significantly addressed by our CSA method. Since our CSA can calculate co-salient attentions of multiple frames, valuable visual information in unlabeled frames can be captured and utilized by our tracking model.

**Amount of annotations**. We set different label sparseness of training datasets for our method, SiamRPN++_CSA_. As shown in Table 6, our method is tolerant of a wide range of annotation sparseness. The tracking performance of our method is still superior to its fully supervised baseline even when the amount of annotations is reduced by 150 times.

**Number of reference frames in CSA**. To explore the impact of our CSA with different numbers of reference frames, we evaluate several CSA variants, shown as Table 7. Numbers of reference frames are denoted as *T*. We observe that our method reaches peak performance at T=3, and the speed does not decrease much. Therefore, T=3 is our default setting in this paper. There are two potential reasons for the performance degradation when T=4. First, high-confidence reference frames often occur close in time, limiting the diversity of target appearances. Second, maintaining a larger number of frames increases the risk of error accumulation, which may negatively affect tracking quality.

**Qualitative analysis.** To demonstrate how our method improves tracking performance over the baseline SiamRPN++, we visualize the response maps and search region features in Figure 8. Specifically, we compare feature maps from layer-5 of the backbone, illustrating three example channels. Our method produces clearer and more target-focused features, with reduced background noise. Different channels capture different aspects of the target, contributing to richer representations. As a result, the classification branch generates more accurate and robust response maps. Figure 8 compares two sequences: “cattle-2” with distractors and “helmet-19” with occlusion. Our method handles both challenges better than the baseline. Overall, the CSA module enhances target features using multi-frame context, leading to more precise localization and improved tracking performance.

## 5. Conclusions

In this paper, to achieve label-efficient but well-performed tracking, we propose a generalized co-saliency attention mechanism, which can be hierarchically integrated into various tracking frameworks and enable the model to be trained in a weakly supervised manner. The proposed method aims to capture the common significant targets in multiple search regions cropped from different frames, from both spatial and channel levels. The target representation in the current search image can be enhanced via the output co-salient attention map. To achieve weakly supervised training, we use sparsely labeled training data, and make full use of remaining unlabeled frames by generating pseudo-labels. To demonstrate the effectiveness and generalization ability of our method, we integrate our CSA module into three different tracking frameworks. Extensive experiments on four general VOT datasets and an egocentric tracking dataset show that our methods can obtain comparable tracking performance to other fully supervised trackers using only one-thirtieth manual annotations for training. Our method also provides a solution for egocentric tracking, which obtains the the new SOTA performance on TREK-150.

### Limitations

During training, extreme fast motion may lead to inaccurate pseudo-labels. While at test time, fast motion or occlusion can cause the target to disappear from the search region, introducing noise into co-saliency learning and increasing tracking failure risk. In the future, incorporating robust target modeling, such as 3D Gaussian distributions or motion-aware interpolation, could enhance reliability and accuracy.

## Figures and Tables

**Figure 1 sensors-25-04691-f001:**
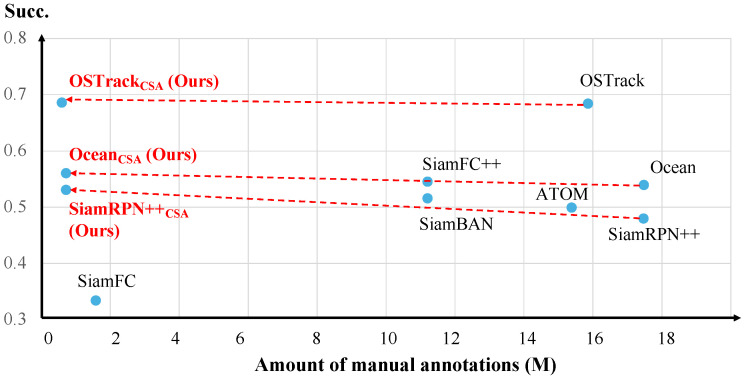
Comparison of our weakly supervised methods with other fully supervised trackers in terms of annotation costs and performance on LaSOT.

**Figure 2 sensors-25-04691-f002:**
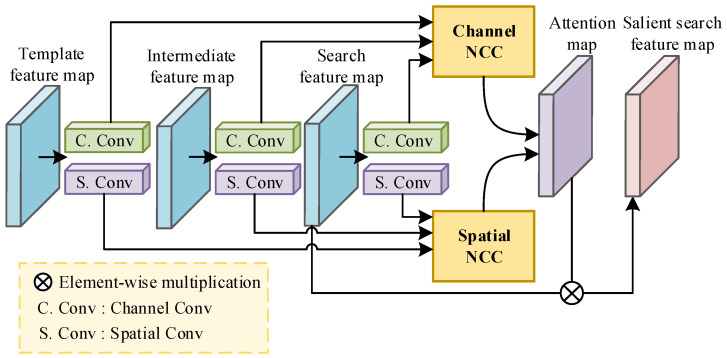
Architecture of the co-saliency learning module. It captures the common salient information of multiple frames in terms of both spatial and channel levels.

**Figure 3 sensors-25-04691-f003:**
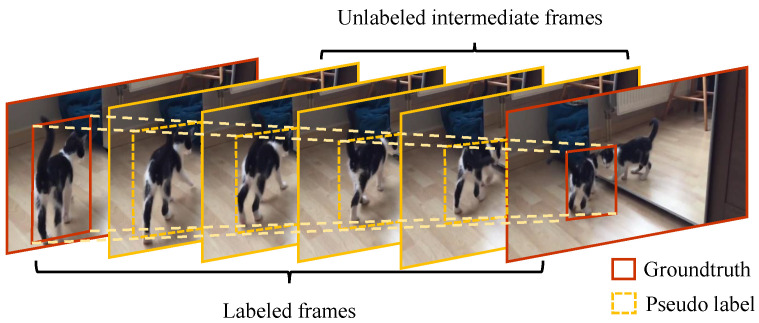
Illustration of our weakly supervised data processing. Pseudo-labels of unlabeled frames are generated based on the groundtruth of the first and last frames from one short fragment in a linear interpolation manner.

**Figure 4 sensors-25-04691-f004:**
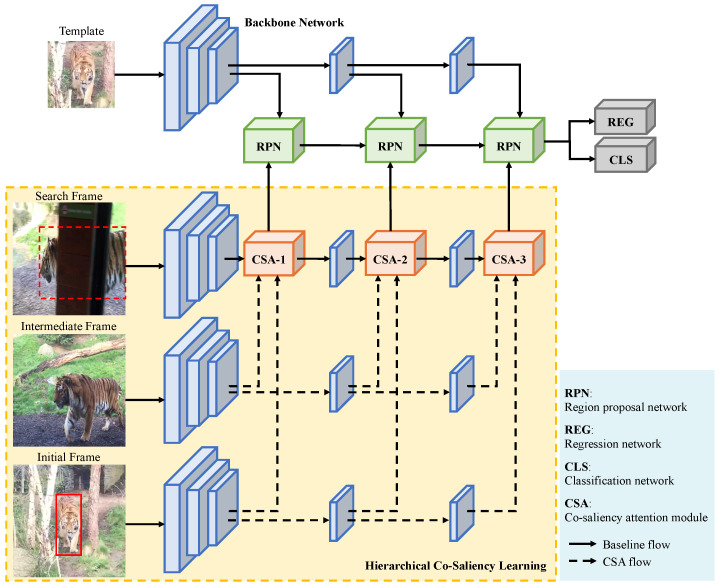
Framework of our CNN-based weakly supervised tracking method. We choose SiamRPN++ as the baseline model. The backbone network, RPN, REG, and CLS blocks maintain the same configures as the baseline.

**Figure 5 sensors-25-04691-f005:**
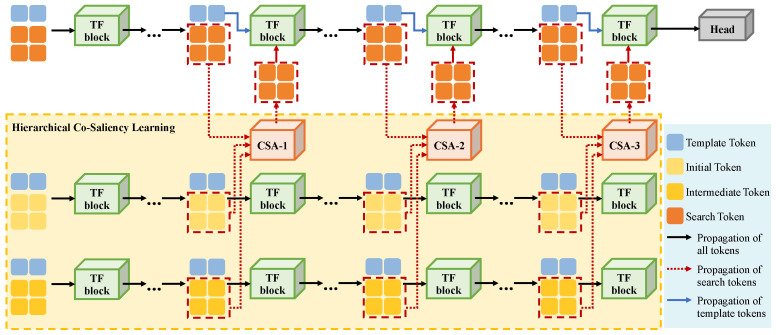
Framework of our Transformer-based weakly supervised tracking method. We choose OSTrack as the baseline model. As the same as the baseline, the model ViT-B [27] consisting of 12 sequential Transformer blocks (TF block) is employed as the backbone network.

**Figure 6 sensors-25-04691-f006:**
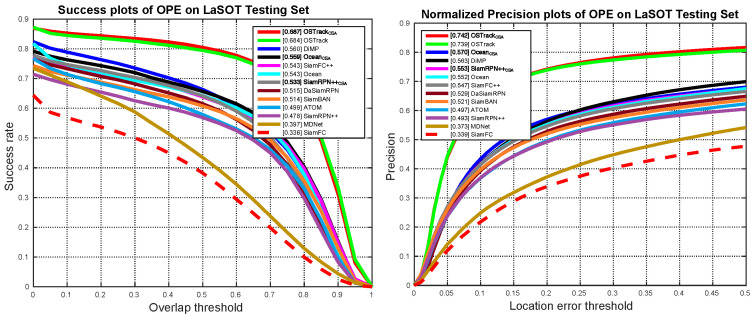
Success and norm precision plots on LaSOT dataset. Our weakly supervised methods obtain comparable results to their fully supervised baselines.

**Figure 7 sensors-25-04691-f007:**
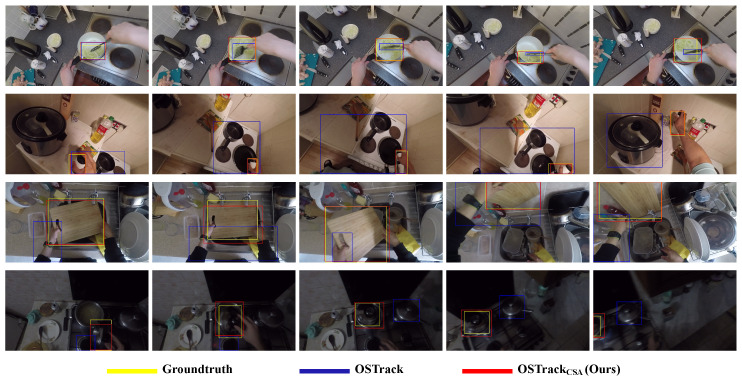
Qualitative comparisons of our method with its fully supervised baseline on TREK-150.

**Figure 8 sensors-25-04691-f008:**
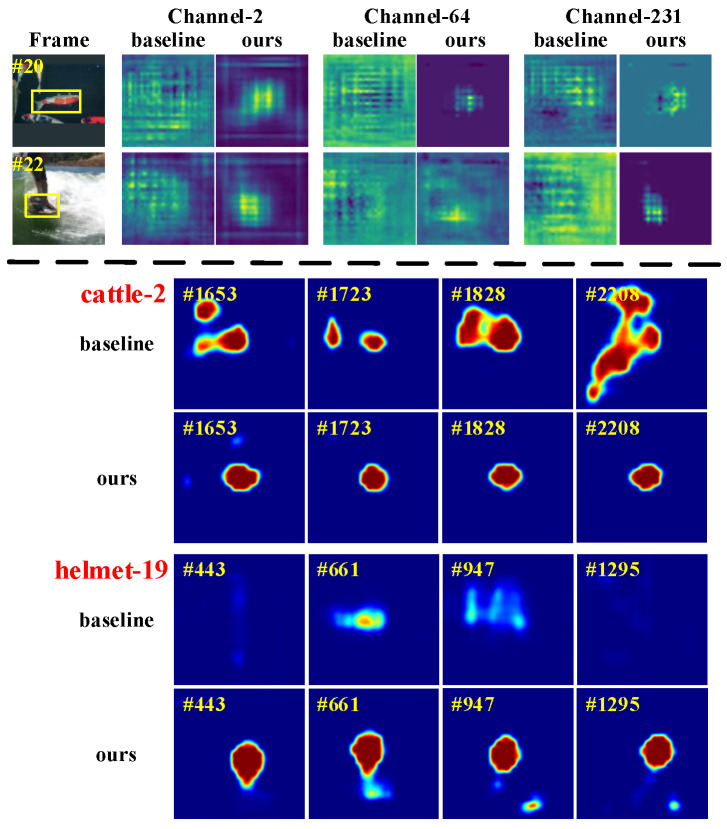
Visualization of search regions’ feature maps and response maps generated from the baseline SiamRPN++ and our method SiamRPN++_CSA_.

**Table 1 sensors-25-04691-t001:** Quality evaluation of the generated pseudo-labels on LaSOT dataset with different sparseness.

Sparseness	30	60	90	150
IoU	0.679	0.575	0.516	0.445
Target ratio	0.968	0.945	0.926	0.907

**Table 2 sensors-25-04691-t002:** Performance comparison on OTB2015 and UAV123. Fully supervised, weakly supervised, and unsupervised trackers are denoted as white, yellow, and pink. The top three results are marked as **red, blue, and green, respectively.**

	OTB2015			UAV123	
	Succ.	Pre.		Succ.	Pre.
MDNet [20]	0.652	0.878	SiamRPN [18]	0.527	0.748
Ocean [16]	0.658	0.863	MDNet [20]	0.528	0.747
ATOM [22]	0.665	0.870	DaSiamRPN [37]	0.586	0.796
DAT [39]	0.668	0.895	Ocean [16]	0.596	0.798
OSTrack [6]	0.675	0.884	SiamRPN++ [5]	0.606	0.797
VITAL [40]	0.682	** 0.917 **	SiamCAR [41]	0.614	0.760
ECO [42]	** 0.691 **	0.910	ATOM [22]	** 0.631 **	** 0.843 **
SiamRPN++ [5]	** 0.693 **	0.897	OSTrack [6]	** 0.678 **	** 0.883 **
RSS [12]	0.675	0.779	RSS [12]	-	-
SS-annotated [11]	0.634	-	SS-annotated [11]	0.521	-
** ${\mathrm{Ocean}}_{\mathrm{CSA}}$ **	0.676	0.900	SiamRPN++_CSA_	0.616	0.819
SiamRPN++_CSA_	0.690	** 0.916 **	** ${\mathrm{Ocean}}_{\mathrm{CSA}}$ **	0.624	0.834
** ${\mathrm{OSTrack}}_{\mathrm{CSA}}$ **	** 0.692 **	** 0.915 **	** ${\mathrm{OSTrack}}_{\mathrm{CSA}}$ **	** 0.677 **	** 0.884 **
ResPUL [10]	0.584	-	ResPUL [10]	-	-
UDT [9]	0.594	0.760	UDT [9]	-	-
LUDT [43]	0.602	0.769	LUDT [9]	-	-
SS-detector [11]	0.619	-	SS-detector [11]	0.519	-

**Table 3 sensors-25-04691-t003:** Comparison of the tracking results and speeds on GOT-10k dataset.

	AO	${\mathrm{SR}}_{0.5}$	${\mathrm{SR}}_{0.75}$	FPS
SiamRPN++ [5]	0.517	0.616	0.325	45
ATOM [22]	0.556	0.634	0.402	30
SiamCAR [41]	0.569	0.670	0.415	52
Ocean [16]	0.590	0.696	0.438	51
SiamFC++ [38]	0.595	0.695	0.479	90
DiMP [21]	0.611	0.717	0.492	43
PrDiMP [23]	** 0.634 **	** 0.738 **	** 0.543 **	30
OSTrack [6]	** 0.717 **	** 0.809 **	** 0.685 **	89
SiamRPN++_CSA_	0.554	0.672	0.369	36
Ocean_CSA_	0.611	0.707	0.480	46
OSTrack_CSA_	** 0.684 **	** 0.789 **	** 0.634 **	65

**Table 4 sensors-25-04691-t004:** Performance comparison on egocentric dataset TREK-150.

	Succ.	Norm Pre.
PrDiMP [23]	0.388	0.402
ECO [42]	0.388	0.413
ATOM [22]	0.400	0.420
TRASFUST [44]	0.403	0.419
LTMU [45]	0.437	0.448
LTMU-F [1]	0.456	0.477
LTMU-H [1]	0.461	0.486
OSTrack [6]	0.502	0.562
OSTrack_CSA_	** 0.538 **	** 0.602 **

**Table 5 sensors-25-04691-t005:** Impact of our CSA module on the tracking performance.

	Ocean			OSTrack		
	Fully S.	Weakly S.	CSA	Fully S.	Weakly S.	CSA
Succ.	0.543	0.532	**0.559**	0.684	0.676	**0.687**
Norm Pre.	0.637	0.628	**0.643**	0.777	0.768	**0.788**
Pre.	0.552	0.551	**0.570**	0.739	0.736	**0.742**

**Table 6 sensors-25-04691-t006:** Impact of the sparseness degree of manual annotations in the training dataset.

Tracker	Succ.	Norm Pre.	Pre.
SiamRPN++^1^	0.478	0.572	0.493
${\mathrm{SiamRPN++}}_{\mathrm{CSA}}^{30}$	**0.533**	**0.637**	**0.553**
${\mathrm{SiamRPN++}}_{\mathrm{CSA}}^{90}$	0.519	0.624	0.538
${\mathrm{SiamRPN++}}_{\mathrm{CSA}}^{150}$	0.497	0.600	0.504

**Table 7 sensors-25-04691-t007:** Impact of the number of reference frames in our CSA.

Tracker	Succ.	Norm Pre.	Pre.	FPS
${\mathrm{SiamRPN++}}_{\mathrm{CSA}}^{T=2}$	0.485	0.584	0.513	38
${\mathrm{SiamRPN++}}_{\mathrm{CSA}}^{T=3}$	**0.533**	**0.637**	**0.553**	36
${\mathrm{SiamRPN++}}_{\mathrm{CSA}}^{T=4}$	0.508	0.606	0.529	34

## Data Availability

Data are contained within the article.

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
