# Peer review of "Generalized Hierarchical Co-Saliency Learning for Label-Efficient Tracking"

_sensors, 2025, doi:10.3390/s25154691_

Round 1
Reviewer 1 Report
Comments and Suggestions for Authors
The manuscript proposes a weakly supervised visual tracking method based on generalized co-saliency learning. The key motivation is to reduce the cost of manual annotations by using co-salient information from unlabeled frames. Experimental results across several benchmarks support the effectiveness of the method.
However, there are several issues that should be addressed.
- The novelty of the CSA module is promising, but the difference from existing weakly/unsupervised tracking methods needs clearer articulation.
- The reliability of pseudo labels generated via linear interpolation is questionable and should be analyzed more thoroughly.
- The ablation study is insufficient and should include evaluations of CSA design choices, such as spatial or channel attention, and different numbers of reference frames.
- The integration of CSA into Transformer-based trackers lacks clarity and requires a more detailed explanation with better visualization.
- How is label accuracy guaranteed when modality information is missing or degraded?
- Consider discussing potential failure cases and providing more qualitative analysis.
- Releasing code or implementation details would greatly improve reproducibility.
Author Response
1. The novelty of the CSA module is promising, but the difference from existing weakly/unsupervised tracking methods needs clearer articulation.
Response: Thanks for your valuable comments. We have added comparisons between our method and existing unsupervised and weakly supervised tracking approaches in Section 2.2 (Label-Efficient Visual Tracking), with the differences highlighted in blue.
2. The reliability of pseudo labels generated via linear interpolation is questionable and should be analyzed more thoroughly.
Response: Thanks for your valuable comments. We have reorganized Section 3.2 (Weakly Supervised Data Processing) to provide a more detailed discussion and analysis of the reliability of pseudo labels generated by linear interpolation, with the revisions highlighted in blue.
3. The ablation study is insufficient and should include evaluations of CSA design choices, such as spatial or channel attention, and different numbers of reference frames.
Response: Thanks for your valuable comments. We provide performance of CSA variants with different numbers of reference frames in Table 7, along with the corresponding analysis in Section 4.4 (Ablation Study), with the relevant content highlighted in blue.
4. The integration of CSA into Transformer-based trackers lacks clarity and requires a more detailed explanation with better visualization.
Response: Thanks for your valuable comments. We have provided a more detailed description in Section 3.4 (Transformer-based Weakly Supervised Tracking) to clearly explain how our CSA module is integrated into Transformer-based tracking frameworks. Revisions are marked in blue.
5. How is label accuracy guaranteed when modality information is missing or degraded?
Response: Since our CSA module does not require precise target annotations but only requires the target to be present within the search regions cropped based on pseudo labels, our method is highly tolerant to pseudo label inaccuracies. In Section 3.2 (Weakly Supervised Data Processing), we provide an in-depth evaluation and analysis of the quality of the pseudo labels. We observe that in over 96.8% of cases, the generated pseudo labels successfully meet the requirement of cropping regions that contain the complete target. Furthermore, as the training process utilizes large-scale data, a small number of inaccurate samples (such as those with missing information) do not significantly impact the overall performance of the model.
6. Consider discussing potential failure cases and providing more qualitative analysis.
Response: Thanks for your valuable comments. We have added a discussion on the limitations of our method at the end of the manuscript, along with potential failure cases, with the relevant content marked in blue. In addition, we have included more qualitative comparisons and detailed analysis to further demonstrate the effectiveness of our CSA module in improving tracking performance (marked in green in Section 4.4 Ablation Study).
7. Releasing code or implementation details would greatly improve reproducibility.
Thanks for your valuable comments. We have added some implementation details in the first paragraph of Section 4 (marked in green), and also attached our code in the supplementary material.

Reviewer 2 Report
Comments and Suggestions for Authors
In the manuscript titled "Generalized Hierarchical Co-Saliency Learning for Label-Efficient Tracking", No. sensors-3741255, authors propose a weakly supervised tracking method based on co-saliency learning, which can be flexibly integrated into various tracking frameworks to reduce annotation costs and further enhance the target representation in current search images. The method has achieved SOTA results compared to recent research. However, I have some concerns outlined below:
- Please open source code to validate its effectiveness.
- Although in most videos, the target’s position and shape do not change significantly within one second, making sparse labeling feasible, there are special cases, such as racing or table tennis videos involving fast motion, where it remains to be clarified how the proposed method handles such scenarios.
- For the issue of object boundary localization, has the author considered addressing it by introducing a 3D Gaussian distribution? Similar approaches include: transforming traffic accident investigations: a virtual-real-fusion framework for intelligent 3D traffic accident reconstruction.
- The authors mention that precise annotation of video frames is costly. In practice, moving away from manual annotation can easily introduce noisy labels, and it is therefore necessary to discuss, e.g., psscl a progressive sample selection framework with contrastive loss designed for noisy labels.
- In the experimental setup section, please list all software tools used and hardware specifications in detail.
- The method description currently lacks pseudocode or an algorithm flowchart.
Author Response
1. Please open source code to validate its effectiveness.
Response: Thanks for your valuable comments. We have attached our code in the supplementary material.
2. Although in most videos, the target’s position and shape do not change significantly within one second, making sparse labeling feasible, there are special cases, such as racing or table tennis videos involving fast motion, where it remains to be clarified how the proposed method handles such scenarios.
Response: Thank you for your valuable comment. Our method does not specifically account for extremely fast-motion scenarios such as racing or table tennis. However, our label quality evaluation (Tab. 1, Section 3.2) on the large-scale LaSOT dataset shows that in over 96% of cases, the pseudo labels generated meet the requirements of our weakly supervised training. Moreover, given the scale of the training data, a small number of inaccurate samples has only a minimal impact on overall model performance. Following your suggestion, we have added an in-depth analysis of the limitations of our method in the revised manuscript (marked as blue).
3. For the issue of object boundary localization, has the author considered addressing it by introducing a 3D Gaussian distribution? Similar approaches include: transforming traffic accident investigations: a virtual-real-fusion framework for intelligent 3D traffic accident reconstruction.
Response: Thank you for the insightful suggestion. We have not incorporated 3D Gaussian distribution in our current framework, as our focus is on developing a flexible weakly supervised tracking method that can be easily integrated into existing 2D tracking pipelines. While 3D Gaussian modeling may offer advantages in precise boundary localization, it typically requires additional assumptions or depth information, which is not available in our setting. Nevertheless, we agree that this is a promising direction for future work. We have added a discussion on this point under the limitations (marked as blue).
4. The authors mention that precise annotation of video frames is costly. In practice, moving away from manual annotation can easily introduce noisy labels, and it is therefore necessary to discuss, e.g., psscl a progressive sample selection framework with contrastive loss designed for noisy labels.
Response: Thanks for your valuable commnets. The input images to CSA are used to generate co-saliency attention maps, which do not require precise target annotations but only need the target to be present within the search regions cropped based on pseudo labels. As a result, CSA inputs from unlabeled frames are highly tolerant to pseudo label inaccuracies. Although linear interpolation is not reliable for modeling complex scenarios such as nonlinear target motion, it is sufficient for our method because precise annotations are not strictly required. To support this claim, we evaluate the quality of the pseudo labels generated on LaSOT dataset using two metrics. Detailed evaluation and analysis can be found in Section 3.2 and Tab. 1.
5. In the experimental setup section, please list all software tools used and hardware specifications in detail.
Response: Our work is implemented in Python using PyTorch. Training is performed using NVIDIA A800 GPUs, while inference is conducted on a single NVIDIA GeForce RTX 2080 Ti GPU. Thanks for your valuable comments, we have added the above information to the revised manuscript (first paragraph in Section 4, highlighted as green).
6. The method description currently lacks pseudocode or an algorithm flowchart.
Response: Thanks for your valuable comments. To provide a clearer explanation of our weakly supervised training pipeline for CSA-enhanced trackers, we have added an algorithm flowchart. The details are described in the first paragraph of Section 3 and are highlighted in blue in the revised manuscript.

Reviewer 3 Report
Comments and Suggestions for Authors
GENERAL COMMENTS
This article proposes a weakly supervised tracking method based on co-saliency learning. This will enable an easier integration into various tracking frameworks, with the aim to reduce annotation costs. However, the approach needs more clarity; I Suggest you give pseudo-codes of the following processes:
- Building of an environment capable of weakly supervised training for tracking frameworks; How Weakly supervised co-saliency Learning is performed (particularly how to extract the common significant foreground areas from a set of images)
- Perform Weakly supervised data processing, CNN-based Weakly Supervised Tracking and Transformer-based Weakly Supervised Tracking
This will help support the narrative given, and ease replicability.
SPECIFIC COMMENTS
- Line 1: Change “…Liu et al. [12] proposes…” to “…Liu et al. [12] propose…”
- Lines 91-92: The following expression is not clear. There is a need to be more explicit, by giving a meaningful narrative on Figure 1 and/or a detailed caption:
“Intuitive comparisons of our methods with other trackers in terms of the performance and annotation costs can be reflected in Fig. 1”
- Lines 132-133: Change “… Liu et al. [12] proposes a reliable…” to “…Liu et al. [12] propose a reliable…”
- Lines 185-186: Change “…represent the mean and standard deviation of the two feature
vectors P and Q…” to “…represent the couples of the mean and standard deviation of the feature vectors P and Q respectively...”
- Line 194: Change “. SAt ∈ R(T−1)HW×H×W.” to “ SAt ∈ R(T−1)×H×W ”
- Line 201: Change “. CA ∈ R(T−1)C×C.” to “ CA ∈ R(T−1)×C×C ”
- Equation (7): You must recall how Conft to and Nt are generated.
Author Response
1. I Suggest you give pseudo-codes of the following processes. This will help support the narrative given, and ease replicability.
Response: Thanks for your valuable comments. To provide a clearer explanation of our weakly supervised training pipeline for CSA-enhanced trackers, we have added an algorithm flowchart. The details are described in the first paragraph of Section 3 and are highlighted in blue in the revised manuscript.
2. Some typo issues.
Response: Thanks for your valuable comments. We have carefully checked and revised our manuscript based on your suggestions.
3. Lines 91-92: The following expression is not clear. There is a need to be more explicit, by giving a meaningful narrative on Figure 1 and/or a detailed caption.
Response: Thanks for your valuable comments. We have revised this sentence to “An intuitive comparison between our weakly supervised methods and corresponding fully supervised baseline trackers is illustrated in Fig.1, emphasizing the balance between annotation efficiency and tracking performance on LaSOT.”.
This revised description provides a clearer and more informative narrative, as highlighted in red in the revised manuscript.
4. Equation (7): You must recall how Conft to and Nt are generated.
Response: Thanks for your valuable comments. We have added a detailed explanation of Conft, marked in red in Section 4.1.

Round 2
Reviewer 1 Report
Comments and Suggestions for Authors
This is a well-executed study worthy of publication. I hope to see more excellent work from the authors in the future.
Reviewer 2 Report
Comments and Suggestions for Authors
all my concerns have been addressed.
Reviewer 3 Report
Comments and Suggestions for Authors
The revised version has addressed all concerns raised.